

# The effect of mangrove restoration on avian assemblages of a coastal lagoon in southern Mexico

Julio Cesar Canales-Delgadillo[1], Rosela Perez-Ceballos[1], Mario Arturo Zaldivar-Jimenez[2], Martin Merino-Ibarra[3], Gabriela Cardoza[4] and Jose-Gilberto Cardoso-Mohedano[1]

[1] Instituto de Ciencias del Mar y Limnología, Universidad Nacional Autónoma de México, Ciudad del Carmen, Campeche, México
[2] ATEC Asesoría Técnica y Estudios Costeros SCP, Mérida, Yucatán, México
[3] Instituto de Ciencias del Mar y Limnología, Universidad Nacional Autónoma de México, Ciudad de México, México
[4] Centro de Investigación de Ciencias Ambientales, Universidad Autónoma del Carmen, Ciudad del Carmen, Campeche, México

Corresponding author
Rosela Perez-Ceballos,
ryperezce@conacyt.mx,
rosela.perezc@gmail.com

## ABSTRACT

**Background**. Mangrove forests provide many ecosystem services, including the provision of habitat that supports avian biodiversity. However, hurricanes can knock down trees, alter hydrologic connectivity, and affect avian habitat. In 1995, Hurricanes Opal and Roxanne destroyed approximately 1,700 ha of mangrove forest in Laguna de Términos, Mexico. Since then, hydrological restoration has been implemented to protect the mangrove forest and its biodiversity.

**Methods**. Since avian communities are often considered biological indicators of ecosystem quality, avian diversity and species relative abundance were evaluated as indicators of mangrove restoration success by comparing undisturbed mangrove patches with those affected by the hurricanes. Using bird surveys, similarity analyses, and generalized linear models, we evaluated the effects of water quality variables and forest structure on the relative abundance and diversity of the avian community in disturbed, restored, and undisturbed mangrove patches.

**Results**. Higher bird species richness and relative abundances were found in disturbed and restored sites compared to the undisturbed site. After restoration, values of frequency of flooding, water temperature, tree density, and the number of tree species were more similar to that of the undisturbed site than to the values of the disturbed one. Such variables influenced the relative abundance of bird guilds in the different habitat conditions. Furthermore, some insectivorous bird species, such as the Yellow Warbler and Tropical Kingbird, were found to be similarly abundant in both undisturbed and restored sites, but absent or very low in occurrence at the disturbed site.

**Conclusions**. Collectively, our results strongly suggest that hydrologic restoration helps to enhance niche availability for different bird guilds, including water and canopy bird species. Our work can help inform management strategies that benefit avian communities in mangrove forests and wetland systems.

## INTRODUCTION

Mangrove forests provide valuable ecosystem services. These highly productive ecosystems can prevent erosion, trap sediments, and provide wind protection for coastal communities (*Warren-Rhodes et al., 2011*). Mangrove forests also support important nursery habitats for a variety of organisms including pelagic and marine vertebrates and invertebrates, mammals, and birds (*Nagelkerken et al., 2008*; *Lee et al., 2014*; *Serafy et al., 2015*). In Mexico, mangroves provide important habitats for specialist species such as the Mangrove Cuckoo (*Coccizus minor*), the Mangrove Finch (*Camarhynchus heliobates*), the Mangrove Hummingbird (*Amazilia buocardi*), and the Mangrove Warbler (*Setophaga petechia bryanti*) (*Polidoro et al., 2010*; *Andrade et al., 2012*; *Gardner et al., 2012*; *Buelow & Sheaves, 2015*). Mangrove forests can also improve water quality (*Wang et al., 2010*), and provide economically important products. Nevertheless, it is estimated that about 35% of the global mangrove cover has been lost, mainly due to forest clearance for fish farming, urbanization, habitat fragmentation, and fuel and timber production (*Valiela, Bowen & York, 2001*; *Alongi, 2008*; *Donato et al., 2011*).

In Mexico, four mangrove species are distributed along the Gulf of Mexico coast: red (*Rhizophora mangle*), black (*Avicennia germinans*), white (*Laguncularia racemosa*), and button (*Conocarpus erectus*) mangroves, covering approximately 742,000 hectares (ha) (*Giri et al., 2011*). The estimated value of mangroves due to fisheries, carbon sequestration, forestry coastal protection, erosion control, water treatment and other environmental, recreational and traditional uses is about $80,000 to $194,000 USD per $ha^{-1}yr^{-1}$ (*Salem & Mercer, 2012*; *Costanza et al., 2014*). Such a value represents up to a total of $59–143 billion USD for the Mexican mangroves. Globally, Mexico ranks fourth in terms of total mangrove area (*Giri et al., 2011*); however, as in other parts of the world, the rate of degradation of mangrove forests in Mexico has been high in the past century. Mangrove cover has been reduced by approximately 10% of the original area in the last 40 years, and about 2% (15,000 ha) of the area that remains has been classified as disturbed (*Valderrama et al., 2014*). In most mangrove forests, there has been a loss of connectivity and a decrease in heterogeneity, which has reduced faunal diversity, including avian populations (*Mohd-Azlan, Noske & Lawes, 2015*; *Hauser et al., 2017*; *Amir, 2018*).

Because restoration can help mitigate climate change effects and the consequential biodiversity loss (*Nilsson & Aradottir, 2013*), interest in the ecological restoration of coastal wetlands is currently growing (*Palmer, Hondula & Koch, 2014*; *Suding et al., 2015*). The disturbance of coastal wetlands is often accompanied by changes in hydrological patterns, including the hydroperiod, which is defined as the amount of time, the frequency, and the level with which a wetland is covered by water, a key factor determining success in wetland restoration (*Turner & Lewis, 1996*; *Zaldívar-Jiménez et al., 2010*; *Wortley, Hero & Howes, 2013*). For instance, sediment deposition in the tidal channels affects the amount of time and the frequency at which a mangrove is flooded (*Woodroffe et al., 2016*). These kinds of changes affect the water quality and the composition of plant and animal communities (*Schaffelke, Mellors & Duke, 2005*; *Krauss et al., 2006*; *Crase et al., 2013*). Hence, the success of ecological restoration can be assessed through a system of indicators that generate
information about the recovery of wetland functions. Such a system might include landscape features, biogeochemical processes (*Cvetkovic & Chow-Fraser, 2011*; *Zhang et al., 2012*), ecosystem services, and the composition of biological assemblages (e.g., vegetation, crustaceans, mollusk, and vertebrates) (*Thornton & Johnstone, 2015*; *Salmo, Tibbetts & Duke, 2016*). Therefore, species richness, abundance, and community structure can be used to evaluate the biological outcomes of restoration efforts (*Zhao et al., 2016*).

Because of their diverse roles within the trophic webs, bird communities are key elements for describing the energetics of ecosystems (*Wenny et al., 2011*; *Adame et al., 2015*; *McFadden, Kauffman & Bhomia, 2016*), and are considered useful indicators of ecosystem health (*Canterbury et al., 2000*; *Bryce, Hughes & Kaufmann, 2002*; *Catterall et al., 2012*). For instance, monitoring of pollutants such as DDT and organochlorines, and the effects of variations in hydroperiod in the Everglades were carried out using water and wading bird species (*Frederick et al., 2009*; *Lavoie et al., 2010*; *Lantz, Gawlik & Cook, 2011*; *Boyle, Dorn & Cook, 2012*). Moreover, bird diversity has been used as an indicator of temporal changes in mangrove health (*Behrouzi-Rad, 2014*), and to assess the impacts of climate change and coastal development (*Ogden et al., 2014*). While avian community monitoring can be a useful tool for evaluating the health of wetland ecosystems, its use for assessment of ecological restoration has been rarely employed (*Weller, 1995*; *Cui et al., 2009*; *Gyurácz, Bánhidi & Csuka, 2011*; *Li et al., 2011*; *Catterall et al., 2012*; *Zou et al., 2014*).

Laguna de Términos is a coastal lagoon located in Mexico along the southwestern coast of the Gulf of Mexico. It is the second largest coastal lagoon in Mexico, supporting approximately 259,000 ha of mangrove forest (33% of all mangrove forest in Mexico) and 262 bird species, among other vertebrate taxa (*Villalobos-Zapata & Mendoza-Vega, 2010*). As a result, Laguna de Términos has been recognized as a Ramsar wetland of international importance (*Chape, Spalding & Jenkins, 2008*). However, due to fisheries, oil-extraction activities, illegal timber exploitation, and urbanization, nearly 26% of the mangrove habitat in this lagoon is considered degraded (*Zaldívar-Jiménez et al., 2017*). Additionally, in 1995, the lagoon was affected by Hurricanes Opal (category 4) and Roxana (category 3), which destroyed 1,700 ha of mangrove (*Pérez-Ceballos et al., 2013*). Fallen trees blocked some creeks (hereafter tidal channels), mainly the secondary ones. The silting of channels altered the hydroperiod patterns and biogeochemical conditions, leading to the mortality of adult trees and inhibiting the natural regeneration of the mangrove. Moreover, this condition has led to a constantly increasing area of dead trees around the points affected by the hurricanes. In order to increase the resilience of these mangrove forests, protect their biological diversity, educate others, and contribute to the sustainable development of the adjacent local communities, from October 2010 to November 2012, restoration activities were implemented. In brief, restoration activities included an environmental and social diagnosis, as well as the formulation of a management plan before the restoration implementation. The primary restoration activity was the desilting and unblocking of the main and secondary tidal channels. Where needed, new secondary channels were dug based on the microtopography analysis of each site selected for restoration. Desilting, unblocking, and channel digging were carried out by local women and men only by hand.

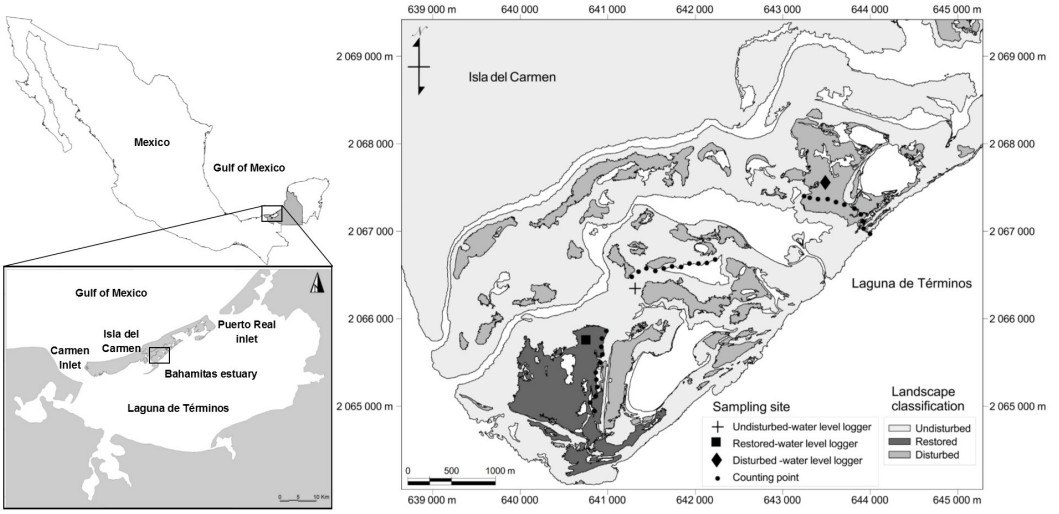

**Figure 1** Location of Bahamitas estuaries with different disturbance levels in Laguna de Términos, Campeche Mexico.

The objective of this study was to use the changes on the avian abundance and diversity as indicators of the restoration success of a mangrove site. We specifically asked: (1) whether the avian community structure differs between not obviously affected mangrove patches and those affected by the hurricanes; (2) whether hydroperiod and water quality influence the diversity or the abundance of birds; and (3) which species could be used as indicator species of post-restoration recovery. The sampling was carried out on a landscape mosaic with different strata: undisturbed patches, patches without restoration, and patches three years after restoration. The water quality variation among the study sites, along with the avian community diversity and abundance of species, were evaluated as indicators of mangrove restoration success.

## MATERIALS & METHODS

### Study area

Laguna de Términos is a coastal wetland located in Campeche, Mexico. It covers about 150 km$^2$ and is connected to the Gulf of Mexico by two inlets at the east and west sides of Isla del Carmen, a calcareous sandbar that supports 5,900 ha of mangrove, of which around 26% are disturbed. The annual average rainfall and temperature are approximately 1,420 mm and 27 ° C respectively (*David & Kjerfve, 1998*). The study area is an estuary known as Bahamitas, located on the east of Isla del Carmen (637787.52E, 2066226.35N, and 633872.60E, 2064181.96N UTM Q15, Fig. 1). The hydrological restoration was implemented from October 2010 to November 2012 through desilting and unblocking of natural tidal channels. Where needed, new secondary channels were created to induce natural regeneration of the vegetation, and to enhance the water quality (oxygenation and salinity) of 1,300 ha of disturbed mangrove through water exchange (*Zaldívar-Jiménez et al., 2017*). The creation of new tidal channels relied on the modeling

of hydrological flow-paths after analysis of a digital elevation model created from the obtained microtopography data. All the work was undertaken by hand using shovels, and involved local inhabitants (10 people ha$^{-1}$), who were trained through workshops on habitat restoration and environmental education, as well as being advised on social and community organization for sustainable development through bird watching and catch-and-release fishing (*Zaldívar-Jiménez et al., 2017*).

## Sampling sites and forest structure

Based on the digital analysis of a Worldview 2013 image and subsequent field surveys, three different patches for sampling were identified and selected according to their condition of degradation: (i) First were undisturbed sites with well-established adult trees and no significant human activities. Here, at least 80% of the trees were alive and there was no evident alteration of the hydrological connectivity due to the effects of hurricanes. (ii) Second were disturbed sites with all the trees dead, except for some few individuals at the edges, and no seedling establishment or with no more than 10% of scrub mangroves alive. (iii) Third were restored sites that showed similar conditions to the degraded site prior to restoration activities implemented during 2010 and 2012 to allow water to flow in and out through the topography of the wetland. After restoration (three years), this site already showed the establishment of seedlings and some saplings, and no more than 50% of the scrub mangroves were dead.

For each sampling site, two 10 m × 10 m random sampling plots were surveyed to determine the forest structure and the number of live trees. On the same plots, measures of diameter at breast height (DBH), canopy height, and basal area were collected to determine the forest cover. Tree density was estimated by counts of all trees with DBH >2.5 cm (*Schaeffer-Novelli, Vale & Cintrón, 2015*). These data were used to investigate the effects of the vegetation structure on bird abundance in each sampling site (*McElhinny et al., 2005*; *Azhar et al., 2013*).

## Hydrologic and water quality parameters

To assess the relationship of bird abundance with the water quality parameters, 11 sampling points were established every 100 m within each survey site to measure the water depth, as well as the temperature and salinity of surficial water of the tidal channels. Water depth was measured using a ruler at the center of the tidal channels. Temperature and salinity were recorded using a parametric probe, YSI-30 (YSI Incorporated, Yellow Springs, OH, USA). The pH and oxidation reduction potential (ORP) were recorded using a portable tester, HI916 (Hanna Instruments, Inc., Woonsocket, RI, USA). All these data were recorded twice monthly. The hydroperiod patterns among the sampling sites were also compared by contrasting the tidal range (the level of the flooding in cm), the flooding duration (the time in hours that a site stand flooded), and the flooding frequency (the number of times per month that a place floods). These measures were recorded during the entire sampling period every 60 min using a HOBO U20-001-01-Ti logger placed at each sampling site.

## Birds survey

To estimate species richness and abundance, monthly bird surveys (December 2014 to June 2015) were conducted by boat. This period of time was selected because it allows finding both migrant birds in winter and residents species in spring and early summer. Counting points ($n = 11$ for each sampling site) were used to record bird species and count individuals per species within each sampling site. We carried out a systematic sampling with a randomly selected start point at each sampling area. Then, the next ten counting points were separated 100 m from each other. Because a count at a particular point can be affected by whether the neighboring points are above or below their averages (*Pendleton, 1995*), we tested for independence of our counting points through a Pearson's test of conditional independence. Additionally, we used the function Moran included in the R package ape (*Paradis, Claude & Strimmer, 2004*) to treat the abundance data across counting points with a Moran's index test to ensure that they were not spatially biased. As recommended for field methods in bird surveys (*Gregory, Gibbons & Donald, 2004*), we used a minimum of four visits to each counting point (four in winter and four in spring). To avoid double count, a single observer counted individual birds at standard intervals of time (10 min). To help deal with varying detectability of different species, observations were made for two consecutive days (*Bibby et al., 1992*; *Gregory, Gibbons & Donald, 2004*; *Schmidt, McIntyre & MacCluskie, 2013*). To deal with species mimicry, besides song and call recognition, birds were identified by a trained observer that sought field marks using $10 \times 42$ binoculars. We recorded all species and individuals seen or heard within the first 20 m radius at each counting point. All species and individuals observed on the vegetation, water, or flying within the observation radius (up to 15 m above) were counted. We did not account for passing or transient birds flying on the sampling areas that did not stop there to feed or rest. The sampling effort was equivalent to approximately 77 h of surveying over the three sampling sites. All bird species observed outside the sampling radius were also recorded but not included in further analyses of species abundance. The reliability of our sampling design concerning species detectability, temporal, and size representativeness was assessed by the implementation of an evaluation framework for ecological research (*Battisti, Dodaro & Franco, 2014*).

## Statistical analyses

The data on hydroperiod and environmental variables were explored for normality through Shapiro–Wilk tests, which are ratios of two estimates of the variance in a normal distribution calculated from a set of observations (*Royston, 1995*). Data that met the assumptions of normality and independence were analyzed using a one-way analysis of variance (ANOVA) to compare the means between the data from all sampling sites. When significant differences were found, ANOVA results were further examined through an honestly-significant-difference Tukey's test, a multiple comparisons procedure to find the differences between all levels of a factor once the hypothesis of equality from the ANOVA test is rejected. When the data did not meet the assumption of normality, we used Kruskal-Wallis one-way analysis of variance by ranks, a non-parametric alternative to ANOVA for multiple comparisons. To contrast the levels of a factor from this analysis, we

 

used the function posthoc.kruskal.nemenyi.test from the R package PMCMR, a post-hoc alternative to performing multiple comparisons for non-parametrics (*Pohlert, 2014*).

Species richness and abundance for each sampling site were calculated based on the number of recorded individuals per species. Diversity was estimated using the inverse of Simpson's Index ($1 \Sigma(n/N)^2$) which indicates greater diversity as the resulting value approaches 1, while the dominance (the extent to which a taxon is more numerous than others) was assessed through the Berger-Parker Index ($d = N_{max}/N_T$), in which the lower the value of d results in a more even dominance in the sample (for details on the formulae see *Ingram (2008)*). To reduce biases caused by non-detected species, the expected species richness was calculated using the Jack1 estimator, which is adequate to estimate actual species richness when the number of sampling units is small (<20 samples or individuals) or when the samples are not the same size. It uses the total number of observed species in a set of samples, the total number of unique species in each sample, and the number of samples for the calculations (*Smith & Van Belle, 1984*; *Gotelli & Colwell, 2011*). Abundance and evenness (how equal the bird community is numerically) among sampling sites were compared through rank-abundance curves. The curves were constructed as implemented in the R package BiodiversityR (*Kindt & Coe, 2005*). From the pooled data, the total number of individuals was calculated to obtain their abundance (*y*-axis) and then ranked from the most to the least abundant species (*x*-axis). Then, the same procedure was implemented for each sampling location (*Kindt & Coe, 2005*). Key bird species to particular sampling sites were identified on the basis of their abundance through a one-way Simper test carried out in PRIMER 7 (*Anderson, Gorley & Clarke, 2008*). The Simper test estimated the contribution of each species abundance to the total dissimilarity among the sampling sites using Bray-Curtis distances, which helps find discriminating features within habitats that explain differences in community composition (*Clarke, 1993*). Additionally, we estimated the indicator species index value (IndVal) to find the value of particular species to each sampling site as indicators of their condition. The IndVal uses both the relative abundance (instead of the absolute abundance), and the relative frequency of each species to estimate its value as a percentage (*Roberts, 2016*). We also investigated the relationship between species abundance and environmental variables by means of redundancy (RDA) and multivariate analyses through generalized linear models (glms). The RDA was selected because there was a linear response of the abundance of birds to the measured environmental variables. For both the RDA and the glms, the birds' abundances were used as the dependent variable, and habitat levels, environmental characteristics, and forest structure were explanatory variables. The RDA test was carried out as implemented in BiodiversityR while the glms were fitted through the R package mvabund (*Wang et al., 2012*). The mvabund functions help test for interactions through multiple testing to predict abundance between sites or treatments. The primary function of mvabund (manyglm) fits a glm to each species in the dataset using a common group of environmental variables. This approach uses a resampling-based hypothesis testing to infer which environmental variables relate to multivariate abundances at community or taxon-specific levels (*Wang et al., 2012*). The independence of the abundance estimates across sampling sites, as well as the quadratic mean–variance and log linearity in the dataset, were checked through

the functions plot and meanvar.plot as implemented in the same package before fitting the models. Several glms were fitted to tests for variable effects. The data set included the mangrove degradation condition as a categorical variable within the environmental matrix. We performed the same analysis but with the data grouped by functional groups according to the birds primary source of food: insectivores, nectarivores, scavengers, macroinvertivores (polychaetes, mollusks, crustaceans), frugivores, omnivores, and meat eaters (fishes, small reptilians, amphibians, and mammals). Because of the difficultly of measuring this directly in the field, this group classification was sourced from literature on local species (*MacKinnon, 2013*; *Fagan & Komar, 2016*), and from specialized web sources (https://www.allaboutbirds.org). Models were ranked according to the Akaike's information criterion (AIC) (*Burnham & Anderson, 2002*; *Burnham, Anderson & Huyvaert, 2011*). All tests were set to be significant at 0.05 level, if not indicated otherwise.

## RESULTS

Pearson's test of conditional independence showed that although the sampling points in each site were closely located, they were independent ($\chi^2 = 26.166, df = 20, p = 0.1604$). Similarly, no evidence of spatial autocorrelation of the abundance was detected across sampling points (Moran's Index value = 0.0074, $p = 0.3335$).

### Water quality parameters and forest structure

The frequency of flooding was approximately two and three times greater at the restored and undisturbed sites, respectively, than at the disturbed one (ANOVA test $F_{2,18} = 12.47$, $p = 0.003$, Fig. S1). However, there were no significant differences among treatments in tidal range or flooding duration (ANOVA test $F_{2,18} = 2.259, p = 0.133$; $F_{2,18} = 2.416, p = 0.118$, respectively).

Environmental variables such as salinity concentration, water depth, pH, and temperature were found to be statistically different among sites (Table 1). The mangrove structure also showed significant differences between sites in terms of the number of tree species, basal area, and tree density (Table 1). However, due to the size of the remaining live trees in the disturbed and restored sites, no statistical differences in the heights of trees between sampled sites were observed (Fig. S2). Finally, the ORP values were not statistically different between treatments. However, the ongoing water exchange due to the greater frequency of flooding in the undisturbed and restored areas might indicate higher oxygen concentration in the water (all ORP values were positive).

### Species diversity and abundance

Fifty-three avian species were recorded in the surveys across the sampling sites (Table S1). One of them, the Reddish Egret (*Egretta rufescens*) is near threatened while the rest are of least concern according to the *IUCN (2016)*. Regionally, five species are under special protection, and one, the Yellow-crowned Night-heron (*Nyctanassa violacea*), is a threatened species (*SEMARNAT, 2010*). For almost half of the recorded species (49%), the populations are trending to increase while 24.5% show a decreasing trend (*IUCN, 2016*).

The total bird species richness was higher in the disturbed and restored than in the undisturbed site, and, for all sampling sites, the observed species richness was lower

**Table 1  Comparison of the mean values of hydroperiod, water-quality parameters and forest structure between studied areas following restoration at the restored site.** Only statistically significant results are shown.

| Variable | Disturbed | Restored | Undisturbed | Test | df | p | Post hoc |
|---|---|---|---|---|---|---|---|
| Flooding frequency (times/month) | 2.71 | 9.00 | 14.42 | $F = 12.47$ | 2.00 | 0. 003 | $R\text{-}D = 0.038$<br>$U\text{-}D < 0.001$ |
| Flooding duration (h) | 588.14 | 404.42 | 314.71 | $F = 2.41$ | 2.00 | 0.118 | – |
| Tidal range (m) | 0.09 | 0.09 | 0.05 | $F = 2.25$ | 2.00 | 0.133 | – |
| Salinity (PSU) | 34.93 | 34.12 | 33.28 | $\chi^2 = 10.25$ | 2.00 | 0. 005 | $U\text{-}D = 0.004$ |
| pH | 8.15 | 8.11 | 8.02 | $F = 12.32$ | 2.30 | 0. 001 | $U\text{-}D < 0.001$<br>$U\text{-}R = 0.023$ |
| Redox potential | 82.09 | 91.18 | 85.18 | $\chi^2 = 3.42$ | 2.00 | 0.180 | – |
| Temperature (°C) | 28.02 | 29.94 | 28.87 | $\chi^2 = 11.81$ | 2.00 | 0. 002 | $R\text{-}D = 0.001$ |
| Depth (cm) | 54.62 | 74.96 | 117.54 | $F = 31.87$ | 2.00 | <0. 001 | $U\text{-}D < 0.001$<br>$U\text{-}R < 0.001$<br>$R\text{-}D = 0.043$ |
| No. of tree species | 1.09 | 1.36 | 1.90 | $\chi^2 = 12.16$ | 2.00 | 0. 002 | $U\text{-}D = 0.008$ |
| Tree density (trees/ha) | 345 | 763 | 709 | $\chi^2 = 6.45$ | 2.00 | 0. 039 | $R\text{-}D = 0.035$ |
| Basal area (m$^2$) | 1.29 | 4.31 | 7.02 | $\chi^2 = 15.39$ | 2.00 | <0. 001 | $U\text{-}D < 0.001$ |

**Notes.**
In the Test column, $F$ is for ANOVA and $\chi^2$ is for Kruskal–Wallis rank sum test. In the Post hoc column, comparisons between pairs of sites (R = restored, D = degraded, U = undisturbed), were tested for significance using either HDS Tukey's or Nemenyi's tests. A dash means that the test was not carried out for that variable.

than expected (Table 2). This result is consistent with our empirical knowledge of the bird community found in the vicinity of our sampling sites. For instance, species such as the migrant Lesser Scaup (*Aythya affinis*), the Northern Pintail (*Anas acuta*), and the American Wigeon (*Anas americana*), as well as the resident mangrove inhabitants, the Muscovy Duck (*Cairina moschata*) and the White-fronted Parrot (*Amazona albifrons*) (both with important population declines in Mexico), were expected but not observed. Diversity indexes showed that there was a clear dominance of few species belonging to the families Ardeidae and Threskiornithidae, especially in the restored site. Even though our sampling effort captured most of the species richness in the study area, the species saturation curve did not reach a flat shape at the end of the surveys (Fig. S3). Accordingly, an increased sampling effort might be necessary to show the true richness of the study area. When pooled data were analyzed, the most abundant species were the Least Sandpiper (*Calidris minutilla*), the Mangrove Warbler, the Black-bellied Whistling Duck (*Dendrocigna autumnalis*), the Green Heron (*Butorides virescens*), the White Ibis (*Eudocimus albus*), and the Great-tailed Grackle (*Quiscalus mexicanus*). Because birds detected in flight were only about 10% of all counts and because there were no statistical differences in the abundance between the whole set of data vs. the data that excluded birds detected in flight (Wilcoxon's test with continuity correction, $W = 1505.5$, $p = 0.7716$), for further analyses, we used the full dataset. Species abundance and evenness were similar for undisturbed and disturbed sites. Although the restored site showed higher bird abundance, this was also the location with the least evenness in species distribution. For instance, higher numbers of species such as the Least Sandpiper and the White Ibis, which depend on mangrove habitats with open or semi-open forest structure, were observed in the restored mangrove patches, while the

**Table 2  Avian community diversity indexes at the study locations.**

| Habitat | n | Observed richness | Mean ± SD | Expected richness | 1-Simpson | Berger parker |
|---|---|---|---|---|---|---|
| Undisturbed | 11 | 28 | 5.27 ± 2.28 | 44.2 | 0.906 | 0.207 |
| Restored | 11 | 41 | 9.00 ± 3.38 | 46.3 | 0.893 | 0.230 |
| Disturbed | 11 | 41 | 9.36 ± 2.20 | 54.5 | 0.945 | 0.133 |

**Notes.**

n, number of sampling points.

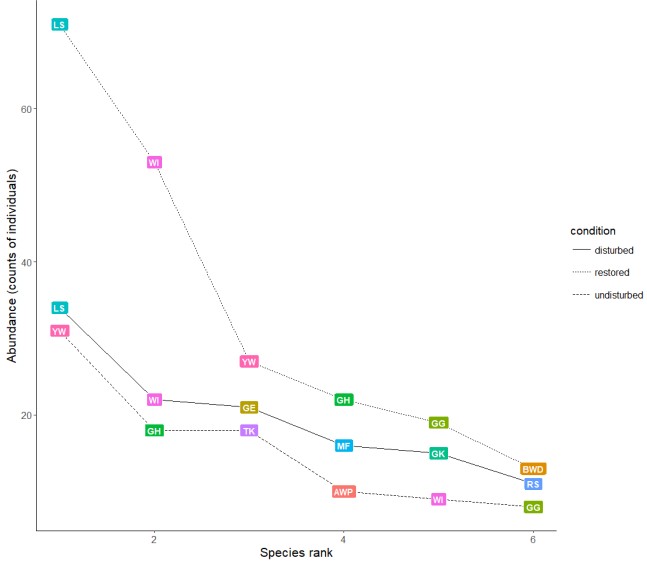

**Figure 2  Ranking of the abundance of bird species by sampling site.** Only the six most abundant bird species for each site are shown: Least Sandpiper (LS), Yellow Warbler (YW), White Ibis (WI), Green Heron (GH), Great Egret (GE), Tropical Kingbird (TK), Great-tailed Grackle (GG), Magnificent Frigatebird (MF), American White pelican (AWP), Black-bellied Whistling Duck (BWD), Green Kingfisher (GK), and Roseate Spoonbill (RS).

Mangrove Warbler, the Tropical Kingbird (*Tyrannus melancholicus*), and the American White Pelican (*Pelecanus erythrorhynchos*) were the least abundant at this site (Fig. 2).

The RDA model was statistically significant (Pseudo-$F_{8,24} = 2.57$, $p = 0.030$), but only a small part of the variation in species abundance within sampling sites was explained by environmental variables since the proportion of unconstrained variance was larger than that of the constrained (0.53 and 0.46, respectively, after 500 permutations for all eigenvalues). When only the first eigenvalue was analyzed, no significant influence of the environmental variables on birds abundance was observed (Fig. S4); however, a trend in some variables such as water temperature, salinity, and tree density to influence the number of birds across sampling sites was detected (Pseudo-$F_{1,24} = 17.97$, $p = 0.055$). To explore the effects of environmental variables on the abundance of birds further, five glms were fitted: one with only environmental variables and not accounting for habitat condition and others that included habitat condition and multiple interactions between environmental variables; however, none of these reached convergence. Thus, the analysis was restricted

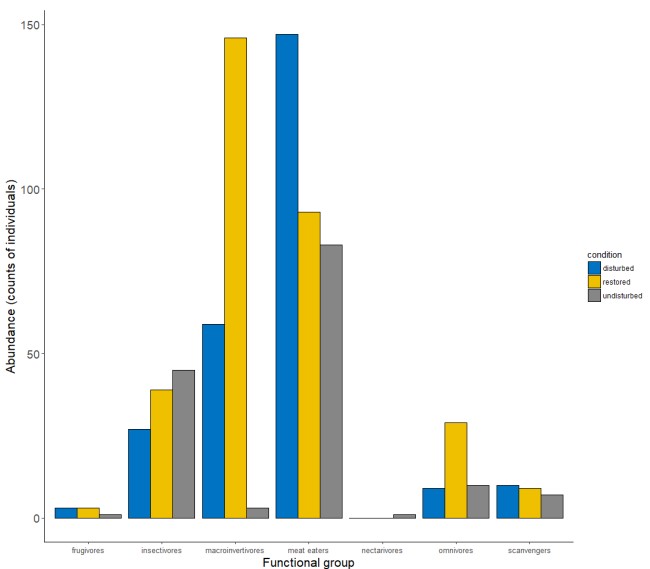

**Figure 3** **Abundance of bird functional groups by site.** In the restored site, the abundance of guilds such as macroinvertivores and insectivores increased as compared with the disturbed site.

to additive models between habitat condition and environmental variables. Because pH and water depth showed high correlation with salinity (Spearman's rank correlation Rho = 0.486, $p = 0.001$; Rho $= -0.595$, $p = 0.001$, respectively), these two variables were dropped from the models. According to the AIC scores, the best models showed significant effects of the environmental variables and forest structure on the abundance of bird species (Table S2). Habitat condition, i. e., the condition of the mangrove as disturbed, restored or undisturbed, showed more substantial effects on the number of detected birds as well as on the distribution of the bird guilds across sites (Fig. 3). Additionally, vegetation cover, as well as the number of tree species, also significantly positively influenced the birds' abundance (Table 3A). When the same models were fitted to each single species, neither environmental variables nor the habitat condition showed significant effects on the estimated abundances ($Padj = 0.730$–$0.900$ for all species and all variables). The multivariate analysis by functional group showed that only detectability significantly influenced the abundance of bird guilds, while the other variables did not show significant effects on the abundance of any of the analyzed functional groups (Table 3B). Similar to the univariate results by species, no significant result was observed when functional groups were analyzed separately ($Padj = 0.172$–$0.932$ for all groups and all variables).

## Key species in mangrove forests

According to the similar percentage analysis, wader bird species, such as the Least Sandpiper, the White Ibis, and the Great Egret (*Ardea alba*) are more associated with the disturbed site. Insectivores such as the Yellow Warbler and the Tropical Kingbird, along with the Green Heron, were more associated with the undisturbed habitat condition. The species associated with the restored site were the Least Sandpiper, the insectivorous Yellow Warbler, and the

**Table 3  Results of the multivariate analysis (best fitted glm).** The greater the deviance, the stronger the effects of the environmental variables on the abundance of bird species (A) and the abundance of functional groups (B).

| Site feature | A | | B | |
|---|---|---|---|---|
| | Deviance | *P* | Deviance | *P* |
| Habitat condition | 162.50 | 0.012 | 22.598 | 0.110 |
| Water temperature | 113.29 | 0.001 | – | – |
| Redox | 82.83 | 0.029 | – | – |
| Water salinity | 71.86 | 0.040 | – | – |
| No. Tree species | 83.02 | 0.003 | 8.623 | 0.388 |
| Basal area | 82.33 | 0.023 | 6.053 | 0.605 |
| Tree density | 76.50 | 0.035 | 6.719 | 0.490 |
| Basal area: Tree density | 45.00 | 0.157 | – | – |
| Detectability | – | – | 17.780 | 0.019 |

**Notes.**
Either dash is for variables not included in the models, or for models not reaching convergence when including those variables.

Green Heron (Table 4). Even though the estimated IndVal value suggested that three bird species might be considered indicators of the different habitat conditions, these results were not significant (Table 5).

## DISCUSSION

The restored tidal channels allowed more frequent water exchange between the Bahamitas estuary and the Laguna de Términos, which increased the frequency of flooding. This, in turn, improved the mangrove soil quality by decreasing the salinity and enabled the natural regeneration of the forest cover. Such changes positively impacted the abundance of insectivorous birds. When comparing the undisturbed and restored sites with the disturbed areas, in the latter, larger duration of flooding favors anaerobic conditions, hindering the oxidation and enabling the rising of soil sulfides concentration (*Reddy & Delaune, 2008*), which affects the ecophysiological functioning of individual plants and is toxic to aquatic fauna (*Lamers et al., 2013*). On the other hand, the lack of connectivity with the main lagoon and the absence of mangrove vegetation, as well as the low water exchange and higher rate of evaporation are factors causing higher pH and salinity conditions (*Tam et al., 2009*; *Molnar et al., 2014*). These conditions are also likely inhibiting the establishment and growth of seedlings and plants. This effect has been observed in other areas of Laguna de Términos (*Agraz-Hernández et al., 2015*). By contrast, the undisturbed site showed more frequent water exchange and dense vegetation cover, leading to slower evaporation rate and lower salinity, as posited by *Lee et al. (2008)*. Although there were not yet statistical dissimilarities between the disturbed and the restored sites, only three years after restoration, the opening and desilting of tidal channels brought down the porewater salinity values (Fig. 4) and other components of water quality in the restored site, being overall more similar to the undisturbed site than to the disturbed one, which probably helped to increase the tree canopy cover, litterfall production and tree growth (*Kathiresan & Bingham, 2001*; *Kathiresan, 2002*; *Polidoro et al., 2010*; *Kamali & Hashim, 2011*). This

**Table 4 Similarity analysis of species abundance among sites.** In (A), the comparison of Undisturbed (UN) vs Restored (RS) is shown, (B) Undisturbed vs Disturbed (DS), and (C) Restored vs Disturbed. Species are ordered by their contribution according to the dissimilarity:standard deviation ratio (D/SD).

**(A)** Av. dissimilarity = 76.83

| | Av.Abund UN | Av.Abund RS | Contribution (%) | D/SD |
|---|---|---|---|---|
| Yellow Warbler | 1.45 | 1.14 | 6.11 | 1.21 |
| Green Heron | 0.89 | 1.08 | 7.35 | 1.12 |
| White Ibis | 0.84 | 0.56 | 5.84 | 1.04 |
| Double-crested Cormorant | 0.31 | 0.70 | 4.72 | 0.96 |
| Tropical Kingbird | 0.61 | 0.09 | 3.72 | 0.89 |
| Great-tailed Grackle | 0.26 | 0.88 | 6.32 | 0.65 |
| Least Sandpiper | 0.09 | 1.14 | 5.55 | 0.58 |

**(B)** Av. dissimilarity = 80.93

| | Av.Abund UN | Av.Abund DS | Contribution (%) | D/SD |
|---|---|---|---|---|
| Yellow Warbler | 1.45 | 0.79 | 6.3 | 1.30 |
| Green Heron | 0.89 | 0.67 | 5.32 | 1.18 |
| White Ibis | 0.84 | 1.03 | 6.32 | 1.15 |
| Great Egret | 0.44 | 0.93 | 5.63 | 1.01 |
| Tropical Kingbird | 0.61 | 0.27 | 3.52 | 1.01 |
| Least Sandpiper | 0.09 | 1.19 | 6.68 | 0.95 |

**(C)** Av. dissimilarity = 75.41

| | Av.Abund RS | Av.Abund DS | Contribution (%) | D/SD |
|---|---|---|---|---|
| Green Heron | 1.08 | 0.67 | 5.02 | 1.27 |
| Yellow Warbler | 1.14 | 0.79 | 4.43 | 1.18 |
| White Ibis | 0.56 | 1.03 | 4.90 | 1.08 |
| Great Egret | 0.62 | 0.93 | 4.84 | 1.08 |
| Yellow-crowned Night-Heron | 0.49 | 0.63 | 3.59 | 1.07 |
| Least Sandpiper | 1.14 | 1.19 | 8.23 | 1.05 |
| Reddish Egret | 0.53 | 0.53 | 3.24 | 1.02 |
| Double-crested Cormorant | 0.70 | 0.34 | 3.59 | 1.00 |

**Table 5 Indicator species index (IndVal) values for key species within each sampling site.** A percentage > 25% and a $p$-value 0.05 mean that the selected species are good indicators for a given habitat condition.

| Species | Habitat condition | IndVal (%) | $p$ |
|---|---|---|---|
| Tropical Kingbird | Undisturbed | 39 | 0.068 |
| Blue-winged Teal | Restored | 29 | 0.057 |
| Mangrove Swallow | Disturbed | 27 | 0.951 |

effect in mangroves has been demonstrated in other sites of the coast of Campeche, where higher rates of litterfall production are associated with lower water salinity (*Chan-Keb et al., 2018*). Besides the fishing and mud-foraging species characteristic of estuarine areas (*Kobza*

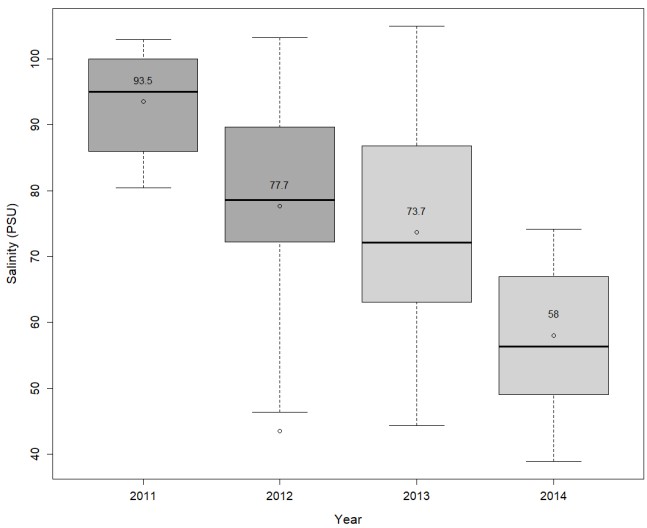

**Figure 4** **The observed changes in porewater salinity though time at the restored site from the beginning to the end of the hydrological restoration (2011–2012), and two years later.** The horizontal lines in the boxes are medians, the open circles are the means, and whiskers are the minimum and maximum values.

*et al., 2004*), higher litterfall productivity, in turn, broadens the spectrum of resources by increasing niches for insectivorous birds' prey species in the restored sites.

Both the disturbed and restored sites showed higher bird abundance and species richness than the undisturbed one. This is similar to the findings reported in structurally complex habitats and island setting, which are found to be more diverse if their habitats show heterogeneity (*Acevedo & Aide, 2008*; *Jones, Marsden & Linsley, 2003*), even if they are disturbed, as were two of our study sites. The results from the models highlight the fact that more heterogeneous habitats such as the restored site and, to some extent, the disturbed one result in more species-rich places, allowing for the existence of more structured bird communities than in the more homogenous undisturbed site (*Báldi, 2008*). This effect is present because high-quality undisturbed mangrove patches surround the disturbed and restored mangrove sites, increasing their heterogeneity. Additionally, while canopy species such as insectivorous birds might use bushes and dead trees from the restored and disturbed sites to feed or rest, bird species that prefer open and semi-open habitats (sandpipers, avocets, cormorants, etc.) are not present in the densely vegetated undisturbed mangrove. They take advantage of the open areas at the disturbed site, which are adequate foraging niches for waders because of the long-lasting floods there. Accordingly, the high abundance of these species could be indicative of sites with impacts affecting the hydroperiod or the vegetation cover. Because of mangrove forests being structurally homogenous when compared to other forest habitats (*Mohd-Azlan, Noske & Lawes, 2015*), the number of bird species recorded was not as high as expected when surveys include other types of tropical forests. This outcome might also be due to habitat preferences of the different bird species recorded, or could be the effect of short-term
monitoring (*Canterbury et al., 2000*). However, the effects of the sampling duration and size might be negligible, since the heterogeneity of the forest structure traits, such as cover and composition, results in a more complex assemblage of bird communities at the landscape level (*Kroll et al., 2014*). Hence, the avian community structure might be more influenced by the heterogeneity, diversity and phenology of the mangrove forest than by the size of the sampled sites (*Mohd-Azlan, Noske & Lawes, 2012*). Moreover, *Chacin et al. (2015)* demonstrated that hydrological fragmentation does not always negatively affect avian abundance since the loss of hydrological connectivity might result in prey concentration, facilitating forage activities for some species of fishing birds. We agree with this hypothesis but only if tidal patterns allow a cyclic interruption-reconnection of the main and secondary tidal channels to allow flooding and drainage of the mangrove sites. The results of the multivariate analysis supported this idea. Moreover, it demonstrated that bird abundance was more influenced by habitat condition (i.e., hydroperiod and forest structure) than by the measured water parameters. Although there were no clear biological effects of the measured water parameters on the abundance of the birds in our study area, they surely determine resource availability because of their effect on the primary producers, and hence on the presence of the benthonic, fish and crustacean communities (*Holguin, Vazquez & Bashan, 2001*; *Sandilyan & Kathiresan, 2015*), on which many bird species feed.

As the characteristics of the habitat influence the distribution of wading species, the sites with less canopy cover provide better foraging areas for waders (*Bancroft, Gawlik & Rutchey, 2002*; *Curado et al., 2013*). Additionally, the selection of open and semi-open areas likely reduces the predation risk and increases the foraging efficiency of wading species (*Pomeroy, 2006*; *Chacin et al., 2015*). Notwithstanding, while tall and broad canopy vegetation might negatively affect the foraging efficiency of water birds, it might become more beneficial for the insectivores (*Tavares & Siciliano, 2013*). Thus, the presence of wading and diving species, together with canopy and undergrowth species using the edges of primary habitat and the emerging vegetation within the restored site, may have influenced the proportions of bird abundance and richness in the study area.

Because our sampling sites are located within a landscape matrix of disturbed and undisturbed patches, the geographical distribution of the study locations likely influenced the number of species found in each sampling site. For example, recordings of birds common to undisturbed sites, such as the Mangrove Warbler, were relatively frequent in the restored and disturbed sites because this species uses features of the disturbed areas as rest spots or merely in the movement between undisturbed patches, as other species do (*Mohd-Azlan, Noske & Lawes, 2015*). This behavior might prevent the identification of particular species from each sampling location just on the basis of observation. Thus, estimates of abundance might be a better indicator. For instance, the abundance of the Mangrove Warbler was higher in both undisturbed and restored areas, although we recorded this species in all studied sites. Additionally, based on the species-by-species abundance, we obtained evidence of non-random use of the restored site as foraging habitats for this and other insectivores (Table 4).

The regrowth of vegetation cover induced by the restoration activities, demonstrated by the increase in the height (0–55 cm) of the mangrove scrub and increased recruitment

(0–79 individualsha$^{-1}$) (see *Echeverría-Ávila et al., 2019*), improved the availability of resources and, hence, the presence and the abundance of insectivorous birds such as the Mangrove Warbler and the Tropical Kingbird. A very close relative of the former, the Yellow Warbler (*Setophaga petechia*), has been considered a key species because of its sensitivity to changes in environmental conditions and specific habitat needs (*Lowther et al., 1999*), and because its populations may change according to the habitat management practices and food availability (*Salgado-Ortiz, Marra & Robertson, 2009*). Since the higher abundance of the Mangrove Warbler was apparently related to the undisturbed areas, it demonstrates its importance as an indicator of habitat with negligible impacts, or of habitats showing signs of recovery. Also, the open and semi-open areas in the restored site contributed to the higher availability of resources for different bird guilds (*Ortega-Alvarez & Lindig-Cisneros, 2012*; *Buelow & Sheaves, 2015*). These areas were more attractive to birds which usually flock in large groups and forage on the ground, mud or inundated areas, leading to higher abundances than in the densely-vegetated undisturbed site. Even though the IndVal was non-significant, both approaches used to identify key species suggested that insectivores better represented the undisturbed site. According to the Simper analysis, the species that best represents the restored mangrove is a meat-eater (the Green Heron), whereas the IndVal suggested that an omnivore (the Blue-winged Teal [Anas discors]) could be an indicator for this site. This lack of coincidence might relate to differing information used by each approach. While Simper uses only the information regarding abundance, IndVal uses the relative abundance and also incorporates the relative frequency of occurrence.

Additionally, the presence of species such as ibises and wood storks is considered evidence of success after the implementation of restoration activities (*Ortega-Alvarez & Lindig-Cisneros, 2012*; *Zhao et al., 2016*). This is because they depend mostly on communities of vertebrates and invertebrates that are sensitive to changes in water and soil quality, induced among other factors, by the alteration of the hydroperiod (*Ogden et al., 2014*). Accordingly, the presence and abundance of the insectivorous and wading bird species similar to those found in the study area are important elements by which to evaluate and monitor the effectiveness of habitat restoration projects in mangrove ecosystems.

## Implications for Conservation

There is no doubt that natural phenomena, such as hurricanes, contribute to habitat heterogeneity. However, severe climatic events may alter the microtopography and hydrological connectivity and make difficult post-hurricane natural regeneration. Such is especially the case in habitats that depend on recurrent flooding and drainage (hydroperiods) such as mangroves. By altering the hydroperiod, dead mangrove areas may increase in size through time. The loss of the vegetation cover and the alteration of the environmental conditions may, in turn, lead to biotic homogenization (*Martínez-Ruiz & Renton, 2018*). After degradation due to loss of hydrological connectivity occurs in mangroves, restoration of water flow through deblocking of main and secondary tidal channels increases the habitat heterogeneity, allowing the resettlement of ecosystem services and strengthening the ecological relationships and structure of the biotic communities living there. It may also improve the hydro-edaphic factors, such as nutrients, water level,

and porewater salinity (*McKee et al., 2002*), from which the micro and macrobenthos, as well as fisheries and birds, are dependent.

Long flooding periods limit the abundance of wading birds, especially the small species foraging on the macrobenthos. Thus, identifying degraded areas and monitoring the hydroperiod, before and after hydrological restoration, will allow for better conservation strategies for the mangroves and their avian communities. As the hydrological connectivity improves mangrove heterogeneity by regenerating vegetation, avian communities can become more diverse, acquiring a higher number of species from different functional groups. Thus, densely vegetated mangroves, together with the restoration of patches unable to regenerate themselves, are essential to maximizing the abundance of specialized bird guilds.

To realize the positive effects of restoration activities, it is necessary to create a link between the restoration programs and the local communities through environmental and economic development education. For instance, birdwatching and sport fishing (catch and release) can be alternatives to socio-economic growth for the benefit of human coastal communities through the development of a green touristic industry in restored places (*Skelton & Allaway, 1996*).

We expect that the restoration activities implemented will increase and maintain the habitat's heterogeneity in the medium and the long term and will strengthen the resilience of the mangrove forests.

## CONCLUSIONS

The restoration activities in the estuaries of Laguna de Términos first helped the natural regeneration of the mangrove forest through the water movement caused by the opening and desilting of tidal channels, making tidal periodicity comparable to that of natural conditions while also reducing water salinity. In addition, water exchange likely favored fish and crustacean production and the appearance of mudflat and shallow water zones, which are attractive foraging areas for different bird guilds. Consequently, habitat heterogeneity and the availability of resources increased, and the avian community became more diverse, especially regarding the abundance of insectivorous birds in the restored areas.

## ACKNOWLEDGEMENTS

We thank Herminia Rejón Salazar and the 'Community of Mangrove Restorers' from Isla Aguada for their support with the field work, Emma Guevara Carrió, Mario Alejandro Gómez Ponce, Hernán Álvarez Guillén and Andrés Reda Deara for their assistance with logistics and field data collection. We thank José Nava from CONANP-Laguna de Términos for the facilities to carry out our surveys.

### Funding

This work was supported by Comisión Nacional Forestal, Gulf of Mexico Large Marine Ecosystem Project, Instituto de Ciencias del Mar y Limnología UNAM, Universidad Autónoma del Carmen (grant number CA-CONAFOR-UNACAR-GOM LME 2014-2016). The funders had no role in study design, data collection and analysis, decision to publish, or preparation of the manuscript.

### Grant Disclosures

The following grant information was disclosed by the authors:
Comisión Nacional Forestal, Gulf of Mexico Large Marine Ecosystem Project.
Instituto de Ciencias del Mar y Limnología UNAM, Universidad Autónoma del Carmen: CA-CONAFOR-UNACAR-GOM LME 2014-2016.

### Competing Interests

The authors declare there are no competing interests. Arturo Zaldívar-Jiménez is a mangrove restoration specialist and manager of ATEC Asesoría Técnica y Estudios Costeros SCP.

### Author Contributions

- Julio Cesar Canales-Delgadillo analyzed the data, prepared figures and/or tables, authored or reviewed drafts of the paper, approved the final draft.
- Rosela Perez-Ceballos conceived and designed the experiments, analyzed the data, contributed reagents/materials/analysis tools, prepared figures and/or tables.
- Mario Arturo Zaldivar-Jimenez conceived and designed the experiments, analyzed the data, contributed reagents/materials/analysis tools.
- Martin Merino-Ibarra authored or reviewed drafts of the paper, approved the final draft.
- Gabriela Cardoza performed the experiments.
- Jose-Gilberto Cardoso-Mohedano authored or reviewed drafts of the paper.

### Data Availability

The raw data is available in the Supplemental Files.

### Supplemental Information

Supplemental information for this article can be found online at http://dx.doi.org/10.7717/peerj.7493#supplemental-information.

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
