# Peer review of "The effect of mangrove restoration on avian assemblages of a coastal lagoon in southern Mexico"

_PeerJ, doi:10.7717/peerj.7493_

## Round 0.1 · original submission · Major Revisions

The reviewers have identified a variety of issues related to presentation of the methods, results, and discussion. I believe most of those issues can be addressed through revisions and author responses. In addition, have some concerns and a number of minor comments / edits as given below:

General
Please check for long sections of continuous text that could be split into separate paragraphs to improve structure.

The standard convention for birds is to capitalize official English common names, for example “Tropical Kingbird”. Check this throughout the manuscript.

Scientific names (Latin names) need to be presented for each species somewhere in the manuscript. Normally, this would be when a species is first mentioned in the text; it might also be done by referring to a Results table that includes species names but there currently is no such table.

Abstract
The abstract should clearly convey categories of mangrove areas used in this study. Differ terms used in the abstract are confusing, including “hydrological restoration” and “mangrove forest restoration” – are they the same? “unaffected mangrove patches” – does it mean undamaged by the hurricane? Please specify. “natural degradation” – what does this mean? Damaged by the hurricane? The term “natural degradation” is not used anywhere else in the manuscript. I recommend succinctly defining three categories: disturbed, undisturbed, and restored, then using these same terms throughout the abstract and in the body of the manuscript.

L122-123 Please specify what you mean by “natural disturbance”. Do you mean hurricane disturbance? Please specify what you mean by “unaffected”. Do you mean not obviously altered by recent hurricanes? Or without major damage? How was it decided if an area was “unaffected”? [reviewer #2 provided related comments]

Methods – The term “site” needs to be clearly defined and consistently used. In some cases there is reference to multiple “restored sites” (L155) while in other cases there is reference to the “restored site” (singular; e.g. L 303). This issue is further confused by problems with the map (Figure 1, see comments below). I am guessing that all samples may have been collected from within a single restored site; if so, authors need use careful wording when making inferences. Comparisons may be presented between the unreplicated restored site and other sites, but since there is (apparently) only a single restoration site, authors need to acknowledge that inferences about effects of restoration may not be valid beyond the specific single site studied. Spatially independent replication is needed to make more general inferences about effects of restoration.

L239 I don’t understand the phrase “Despite the salinity concentrations”. Do you mean “Despite similar salinity concentrations”?

L249 I think a table should be referred to in this sentence listing the 53 species that were found in this study and the mangrove categories where they were found. It could be a Supplemental table if it is too cumbersome put it in the main paper.

L254 This statement should reference the data (e.g. a table or graph) and it would be best to tell us the numbers in this sentence because this is one of the main questions that the study sought to address.

L256-257 Of course this is am obvious mathematical inevitability. I think this sentence should be removed. You could instead tell us how total observed richness compared to expected richness, especially since total richness is not given in Table 2.

L257-261 – The Jack 1 estimate is based on theory and is not based on knowledge of any specific named species. A finding that the Jack 1 estimate is higher than observed total richness simply tells you that further sampling will theoretically yield more species. If you want to name specific species, I think wording should be something like “Higher estimated than observed richness is consistent with our empirical knowledge of the bird community found in the vicinity of our sites; for example, species x,y,z were expected but not observed.”

L262 The curve does not “flatten” (see comment on figure 3 below; Reviewer #1 pointed out this same issue); it is obvious that more sampling will add more species based on the curve, and this is consistent with the Jack 1 estimate being higher than the observed total richness. Consistency in findings between different estimators (rarefaction and Jack 1) is expected – both tell you that more samples would likely yield more species; this is to be expected, especially in the diverse tropics. What concerns me most is that the abstract states that alll three mangrove categories had similar bird richness, but that is contradicted by figure 3b: clearly, richness in the undisturbed is consistently lower than the other two categories for any level of sampling.

L263-264 “These findings indicated that increased sampling is likely necessary to show true richness of the undisturbed mangrove patches...” – it is not only true of the “undisturbed” patches it is true of ALL patch types.

L291-292 – Rather than fitting to individual species, which could have lower power and could be seen as “data snooping” because of the large number of tests made with no a priori hypothesis, I wonder instead about whether there could be associations with bird functional groups? You could make a priori hypotheses about abundance of bird functional groups based on their feeding requirements / preferences.

L 297 What is the difference between disturbed and highly disturbed?

L303 For consistency with the previous paragraph, should this say “restored patches” or maybe “patches” should be changed to “sampling plots” in both paragraphs?

L326 “Although there are not” to “Although there were not”
L332 “mangrove” to “mangroves”
L338 This sentence implies that there is only a single undisturbed site. It needs to be clarified whether multiple sites were sampled or a single site was sampled, and the terms “site” and “area” should be clearly defined. If they are the same, then using just one term would help avoid confusion.
L339 “islands setting” to “island setting”
L341 “de models”??
L342 “in some extent” to “to some extent”
L345 “forests are” to “forests being”
L352 This would only be true after sampling a certain threshold area size related to the scale of forest heterogenity.
L357 “the inner of the mangrove sites” – what does it mean?
L361 “abundance of the birds community” – abundance may be used in reference to a population or count but not a community.
L365 “habitat influence” to “habitat that influence”
L369 “big-sized” to “broad”
L375 Is this statement true? The map seems to show one restored “patch”.
L381 “hamper to define” to “prevent the identification of”
L391 “needs of habitat” to “ habitat needs”
L408 “environmental services” what does it mean? Ecosystem services?
L409 “ living in there” to “ living there”
L416 What do you mean by “more structured”? More diverse? With more functional groups? Or does it mean greater differences from neighboring areas (spatial heterogenity)?
L421 “ birds watching” to “ bird watching”
L423 “of green” to “of a green”
L424 “implemented, contribute to increase” to “implemented will increase”
L425 “long-term to strengthen” to “long-term will strengthen”
L434 “favored” to “likely favored” [do data were presented for this so wording needs to be softened]
L437 “ more structured” – again, what does it mean? Please expand to explain or re-word.


Figure 1 – The Methods (line 168) say that 11 counting points were established for each of the three site types, but on this figure I can see only one string of 11 counting points (most are in “restored” but strangely 3 of them are in undisturbed next to restored). I see an odd letter “u” on the map but the diamond representing the disturbed water logger is missing from the map. Authors should carefully check the formatting of the map. I am guessing that the 11 sampling points displayed are not correctly positioned while the remaining 22 sampling points are missing. It could be more effective to plot counting points as hollow white circles with black edges, as they would show up better on all backgrounds.

Figure 3 – none of these figures have converged to an asymptote. Adding more sites will obviously add plenty more species based on extension of the curve – the curve is not “flattened” (horizontal) near 30 sites. The font size for the axis labels and numbers seems too small on this figure.

Figure 4 – Axis label wording should be capitalized.

Reviewer 1 ·

Basic reporting

This article uses sufficient literature that provided adequate background on mangrove ecology and birds. The figures appears sound except for Figure 1 that may benefit some level of modification. Figure 1 The sampling points in three different habitat treatment is not clearly visible.

Experimental design

The research question has been clearly stated in this study. The knowledge gap in this field of study has also been highlighted and this study is probably the first to attempt to provide some light into the restoration ecology on mangrove birds.

However the sampling techniques used may have created bias in bird observation For instance the sampling location were closely located (100m apart), this in turn would result in double counting and inflation in relative abundance. The use of bird observation to indicate abundance can also be misleading, This is should be treated as relative abundance? The sampling saturation from Figure 3 also do not appears to achieve sampling saturation as being cited in the figure caption.

Validity of the findings

The data collected may need to be reviewed carefully. Justification are need from the authors on how potential bias are dealt with.

Additional comments

General Comments

This work should be commended as it reveals the complexity of habitat restoration in relation to avian assemblages. This work will add to the growing literature on restoration ecology and mangrove birds which are lack in many parts of the world. However there are several areas of concern that may need critical justification prior to publication.

Line 48 vertebratesand 


180 Additional details on the bird survey is needed. No of observers? How multiple observer bias was overcome? The steps taken to reduce misidentification through mimicry and how birds were identified?

186 sampling point were separated by only 100m- This raises concerns on the possibility of double counting and biases. Please provide details and additional justification. The sampling points are relatively close between treatment habitats which may violate the independence of the point count. Such spatially aggregated placement of sampling point causes the spatial-correlation of abundance of target species and double counting probability.

191 birds that were flying above 15 was also included in the analysis. If the study is comparing the utilization of the different types of mangrove patches by the birds-why would the authors, then consider including transient birds or passing by birds in the analysis. Analysis excluding these species may show a different conclusion?

203 & 215 How was the number of individuals determined for each species? No of observation do not represent number of individuals. This can be misleading as the same individual could be recounted over the sampling period.

223 Please provide justification why RDA was used instead of other widely used multivariate analysis in understanding avian ecology?

243 It is unclear why there are no significant differences on the tree heights between degraded, restored and undisturbed areas.

257 Why was the migratory species included in the analysis? These seasonal birds may have a wide distribution and habitat. Please provide justification why not only Passerines were used in the analysis.

341 de models 


In the discussion please provide additional arguments why there are fewer species in the undisturbed forest

The sampling in restored habitat include a wide range of restored period (2-6 years)- Would the heterogeneity between these treatment years have any influence on the avian relative abundance and richness? Please provide some arguments in the discussion.

Figure 1 The sampling points in three different habitat treatment is not clearly visible.
Figure 3 Graphs do not indicate sampling saturation. Please provide details

Reviewer 2 ·

Basic reporting

This manuscript was generally well-written, sufficiently referenced, self-contained, and appropriately structured. However, some attention to minor grammatical errors is required throughout the manuscript: e.g. line 156, 341. Also, individual plots in Figures 2 and 3 should be distinguished as a), b), c)...etc.

Experimental design

The research is within the aims and scope of PeerJ, and its contribution to the literature is welcomed at a time when evaluation of restoration success is needed. While the study design is appropriate, some clarification of the research question is required, and the methods used to analyse the data require further description and justification (see general comments below). Furthermore, results of several of the tests used are not adequately reported, and figures/tables are not consistently referred to in the results section.

When justifying the analytical approach used, I would suggest authors consider the following paper, which is more straightforward in its evaluation of bird community structure and identification of indicator species using the ‘IndVal’ method.
J. Mohd-Azlan, R. A. Noske & M.J. Lawes (2012) Avian species-assemblage structure and indicator bird species of mangroves in the Australian monsoon tropics, Emu - Austral Ornithology, 112:4, 287-297, DOI: 10.1071/MU12018

Validity of the findings

Throughout the discussion, care needs to be taken to ensure the claims do not over reach the data collected. For example, lines 387-389 claim that vegetation regrowth due to restoration increases the availability of resources for insectivorous birds. However, insect availability was not measured, and so increases in insectivore resource availability can only be suggested as a speculative statement.

Additional comments

Lines 121-129: The study questions require some clarification. What is ‘natural disturbance’ in the context of this study (line 122)? Does it include mangrove degradation due to cyclone damage and exclude human-caused disturbance, such as harvesting? Results are displayed ‘disturbed’, ‘restored’, and ‘undisturbed’ categories, and these should be referred to in the study questions.

Line 168: What was the distance between the eleven sampling points?

Lines 202-232: It seems that ANOVAs or Kruskal-Wallis tests were used to determine if there were differences between water quality and vegetation parameters between study locations. There needs to be description of these tests and how assumptions were met in the Methods section. Also, there was no mention of any post-hoc analyses for differences between groups following ANOVAs. Was this completed, and if not, why not?

Line 203 to 212: I would suggest providing better explanations of community structure measures used, for example readers may not be familiar with the meanings of ‘dominance’ and ‘evenness’.

Line 240-241: This section is only reporting on water quality and forest structure results, so I suggest removing ‘we did not find a clear effect of these differences on the bird abundances among the sampled areas’

Line 242: Why aren’t measures of mangrove structure also included Figure 2?

Lines 235-246: Figure 2, Table 1, and the reporting of ANOVA results in the results text is somewhat redundant. I would suggest making Figure 2 more comprehensive (to include mangrove structure) or, alternatively, just use Table 1 to report these results.

Lines 245-256: Suggestions should not be included in the results section, only in the discussion.

Line 254: A figure should be referred to with this statement.

Lines 266-267: The results statement requires reference to a table or figure that reports these results.

Lines 278-280: It would be useful to provide a figure of the RDA so that readers can see these trends.

Lines 282-287: To better justify the steps taken during the glm analysis, I would suggest a better description of the ‘mvabund’ glm process and its assumptions in the methods section.

Line 315: It was not clear from reading the results that insectivourous bird abundance was influenced by tidal channel water quality. Foraging guild was not included in the analysis.

Figure 1. An inset map showing the location of the study location in the Gulf Of Mexico would be useful for the reader not familiar with the geographic location of Laguna de Terminos.

Figure 2. Letters should be used to distinguish individual plots, e.g. a, b, c, d…

Figure 3. The two plots should be distinguished as a) and b) and described as such in the figure caption. Furthermore, the caption should not report results (i.e. ‘ most species were recorded….species richness might be higher…’).

Figure 4. Why are only the first six sites ranked?

Figure 5. At the study site? Need to make clear that this pooled over the restored site only, the disturbed and undisturbed sites are excluded.

Table 1. This is repetitive of Figure 2. Suggest including standard error in the table and placing boxplots for all variables in an appendix and/or supplementary materials, or vice versa.

Reviewer 3 ·

Basic reporting

No comment

Experimental design

Authors used the same methodological approach to detect birds from very different guilds (seabirds, herons, shorebirds, passerines, ...) that are subjected to enormous differences in detectability. While they recognize it (by quoting Bibby et al. 1992) authors should account for it in their models (see e.g. Gibbons & Gregory 2006. Birds IN Ecological census techniques W.J.Sutherland, ed. Cambridge Univ. Press). Without accounting for differences in detectability, both among guilds and between treatments, results of bird abundance (and therefore richness and diversity indexes) between different treatments can be in my opinion strongly biased. One potencial evidence of this bias is that authors had only found 53 bird species troughout the study period in a tropical mangrove area, which overall supports a much higher bird (species) richness. Rarefaction curve is my my opinion a result of potential species detectability with the method used rather than because actual species richness.

Validity of the findings

Based on reported issues of methodological approach used I think that findings are not valid as they are currently stated in the MS. I suggest authors to:

1. Focus response variables (i.e. bird guilds) on the main purpose of the treatment (i..e mangrove restoration). Based on the methodological approach reported, I suggest to focus on waterbirds (i.e. mainly Pelecaniformes, Anseriformes and Charadriiformes) and include the tidal variability in your models. There can be are huge differences in waterbird use of studied areas depending on tidal period (i.e. high and low) and amplitude (see e.g. Basso et al. 2018 Hydrobiologia).

2. If authors would like to include passerine birds, I encourage them to replicate the surveys in the following years increasing the sampling effort to detect least abundant -but important components of the communities- songbirds at mangrove forests. If possible, I woud suggest to develop bird captures using mist nets at each treatment site to account for differences in songbird communities (e.g. Arizaga et al. 2011 Bird Study) at different sites.

Minor comments

Line 254. This result is not in accordance with detailed results (Table 2 and Fig 3).

Line 289. Include the results of model selection based on AIC scores.

Line 296. Remove this sentence.

Lines 296-304. I suggest to move several sentences to the Discussion section

Line 311. Focus your first sentence in the Discussion section to your main findings.

Line 406. Remove this sentence.

Line 426. This is too speculative.

Line 430. Focus your first sentence in the Conclusions section to your own (and not from other quoted studies) findings.

Line 438. This is too speculative.

Additional comments

(While I´m not a native english) In my opinion, the MS is not properly written. Authors used several sentences that appeared basic translations from spanish rather than scientific english writing. With regard to specific terms accounting to (bird) community structure and composition (i.e. abundance -even usied in the title-, richness, diversity), I suggest authors to thoroughly revise the text to homogenize. Specifically about birds, you should properly revise the use of specific terms such as shorebirds, waders, wading birds in order to not confuse the reader (e.g. shorebirds and waders are synonyms, one is used within the Nearctic and Neotropic, and the other is mainly used in the Palearctic). I suggest the same for naming the experimental groups (i.e. disturbed and degraded for the same treatment). There are even some specific words that I can not properly understood (e.g. porewater, litterfall production and gleaning birds...).

---

## Round 0.2 · Major Revisions

The manuscript has been improved, however changes made have led to further questions (as pointed out by the reviewer). In addition, various English corrections along with several more substantial issues and questions as summarized below:

Abstract – The Results section of the Abstract doesn’t adequately summarize the study’s findings. No information is summarized on environmental variables or forest structure. Did they seem to relate to the bird communities? The Discussion repeatedly refers to insectivores – if that was a key finding, it should be mentioned here and in the main Results.

Why are in-text citations of author-date italicized?

L 26 on the local biodiversity -- on local biodiversity

L 27 For clarity, please reword “.. we compared a restored mangrove patch to an unrestored (disturbed) hurricane-affected patch and an undisturbed mangrove patch”

L33 By definition, relative abundances are standardized by the total abundance at each site, so it does not seem possible for relative abundance to differ across sites.

L 36 The similarity percentage analyses of avian community structure showed – Avian community structure analyses showed

L 41 strategies, that benefit – strategies that benefit

L 122 of mangroves sites – of a mangrove site

L 129 study sites, the avian community – study sites, along with the avian community

L 143 Incomplete sentence

L 158 no seedlings establishment – no seedling establishment

L 162 showed already – already showed

L 202 calls recognition – call recognition

L 210 the species detectability – species detectability

L 229 which indicate – which indicates

L 243 is implemented – was implemented

L 250 as indicator of – as indicators of

L 252 value as percentage – value as a percentage

L267 fitted to tests for variables effects – fitted to test for variable effects

L279 Do you mean “sampling points”? My understanding is that “sites” refers to the different types – undisturbed, disturbed, restored. L 277 refers to “points” not sites.

L 292 The ORP – the ORP

L 303 in disturbed and restored – in the disturbed and restored sites

L 310 of few species (Ardeidae and – by a few species (in the families Ardeidae and

L 319 – all birds can fly here, please clarify wording, do you mean “birds detected in flight”?

L 321 original dataset (?) Do you mean the full dataset including birds detected in flight?

L322 showed the higher bird – showed higher bird

L324 the higher numbers – higher numbers

L 338 and no accounting – and not accounting

L 346 showed the more substantial effects – showed more substantial effects
[this sounds like an important finding; please explain the nature of these “substantial effects” ]

L 347 also influenced the bird [influenced in what way? Positive correlation? Negative correlation? ]

L 351 showed that water salinity and detectability influenced significant the relative abundance [what is detectability? If this is a measure of birds it should not be in the same model with a water quality measurement.] Relative abundance of what? Furthermore, in what way did salinity influence abundance? The sentence is missing crucial information as currently written.

L 357 In what way did salinity influence macroinvertivores? Positive correlation, with increased salinity, more macroinvertivores? It needs to be explained whenever a statistically significant result is presented.

L361 best discriminated between disturbed and undisturbed sites [this wording is not informative. We need to know which bird is associated with which specific habitat condition]

L 366 although the significance was marginal. [what does this mean; be specific. Generally if the p-value does not reach alpha, the conclusion must be that there is no association; if you go against this rule then you should explain why you think it is important to point out a non-significant result]

L 370 which improved water and soil quality [in what way? Be specific and refer only to statistically significant differences]

L 372 impacted the relative abundance of insectivorous birds. [this claim is bizarre given that the entire Results section never once mentioned insectivorous birds; this needs to be rewritten]

L 378 are the factors – are factors

L 379 These conditions are also inhibiting the establishment – These conditions are also likely inhibiting the establishment

L 382 a constant water exchange – more frequent water exchange

L 398 as two of – as were two of

L 401 Incomplete sentence

L412 not high as expected – not as high as expected

L 426 abundance was more influenced by habitat condition [more than what?}

L 439 influenced – may have influenced
L 454 increase in the height of the mangrove scrub [no data on height are given in table 1; statements need to be supported by data]

L 465 Incomplete sentence

l 467 coincided that – found that

L 468 the IndVal was marginal [what does it mean?]

L 469 the higher availability – the presumed higher availability

L 469-470 Due to the higher availability of resources for different bird guilds, no match between both methods resulted for the restored site, [I didn’t understand this logic]

L472 what does “marginally mean” - if the alpha criterion is not met, the conclusion should be that there is no pattern for discussion unless a convincing argument is made that a pattern should be considered due to special circumstances.

L 489 such as nutrients, --[nutrient data were not presented; wording needs to change. It could be something like “It may also improve….” - this kind of wording is required because it does not necessarily apply to your site.]

L 495 more diverse because of a higher – more diverse, acquiring a higher

L 498 To keep the effects of restoration activities is necessary – To realize positive effects of restoration activities, it is necessary

L485-505 I have a concern about this concluding paragraph – it seems obvious that hurricanes are a natural cause of habitat heterogenity, and so in order to maintain habitat heterogenity, we NEED HURRICANE DAMAGED MANGROVE PATCHES. This basic issue has not been addressed in any way. If we quickly restored all of the naturally damaged areas, the mangrove would become less heterogeneous, not more heterogeneous. Is there instead an optimal balance between active restoration and maintaining patches that are naturally disturbed by hurricanes? Or is it your position that humans should attempt to overpower nature and quickly remove (restore) the hurricane damaged patches? I don’t recommend an extended philosophical discussion, however, I do believe that it is important that you acknowledge the role (and benefit) of hurricane damage in creating habitat heterogeneity.

L 514 and avian community – and the avian community

Table 1 – There is no way to determine which site pairs are statistically significant. Standard practice would be to utilize symbols such as superscript letters to designate significant differences (e.g. differing letters indicate significant differences). The current Discussion makes statements that are not necessarily supported by results in this table because significant differences between specific sites are not designated. Also, the Discussion refers to decreasing oxygen which is not shown in the table. I think mean values should be given for all measured environmental variables even if they were not statistically different, since all of the variables help give a picture of the conditions at the sites.

Table 3 What is detectability? It does not fit with the other variables.

Figure 2 - I am wondering why this is included as a main figure rather than a supplemental figure? This type of figure is used as a diagnostic but is not useful for understanding differences among sites except that figure B shows the disturbed site has lower richness, but that is already shown in the corresponding table. I don't recommend this as a main figure.

Figure 3 – I don’t see the value of this figure in addressing the questions / hypotheses presented in the paper. The line containing species names is for all habitats combined whereas this paper is supposed to be asking how do avian communities differ among the three habitat types? How does this graph address the main questions? Wouldn’t it be clearer to present the community composition of the different sampling points using a PCA or MDS or similar X,Y coordinate presentation of multivariate axes? That would allow simple visualization of difference (or lack of differences) among communities detected at the 33 sampling points, and the degree of clustering among points from the same habitat type could be easily visualized.

Figure 4 What are the units of salinity on this graph? I don’t recognize “ups”

Reviewer 2 ·

Basic reporting

Although improvement has been made since the first manuscript draft, grammar throughout the manuscript requires a careful edit. Also, please be consistent with terminology throughout the manuscript. For example, in Figure 1 the survey locations are called ‘count points’ but in Figure 2B they are called ‘sampling sites’.

The literature is not sufficiently referenced through-out the introduction, and specific comments highlighting this have been made in the 'General comments for the author'.

The article could be improved by clearly defining the objective of the study and ensuring that analyses used are relevant and not redundant (see specific comments in 'General comments for the author').

Experimental design

There are potential confounding factors that should be addressed given the location of the counting points (Figure 1). For example, could differences in bird species abundance and diversity between undisturbed, restored, and disturbed forests be due to edge effects (i.e. from Figure 1 it looks like most count points for restored forests are located primarily along the forest edge, while disturbed count points are located primarily inside the forest). Additionally, some of the undisturbed count points are very close (<100m) to patches of disturbed forest – is it possible that this could have influenced species recorded in the undisturbed forest?

Validity of the findings

The main conclusion of the paper is not well supported by the results as displayed in figures, please see specific comment #17 in 'General comments for the author'.

Additional comments

1. Lines 1-2: Given the study isn’t focused solely on changes in avian abundance, I would suggest a slight title change: ‘The effect of mangrove restoration on avian assemblages of a coastal lagoon in southern Mexico’ or ‘The effect of mangrove restoration on avian diversity and abundance in a coastal lagoon in southern Mexico’.

2. Lines 21-22: It is perhaps more accurate to say that ‘mangrove forests provide many ecosystem services, including the provision of habitat that supports avian biodiversity’.

3. Lines 26-29: These two sentences are indicative of a lack of consistency in study objectives throughout the article. Was the study objective to determine the influence of hydrological restoration on biodiversity, or was the objective to use avian diversity and abundance as indicators of restoration success? This needs to be established early on in the article, and be discussed consistently throughout the manuscript.

4. Line 31: Should ‘relative abundance’ be ‘relative abundance and diversity’?

5. Lines 34-35: I suggest the following: ‘Furthermore some bird species, such as the Yellow Warbler and Tropical Kingbird, were found to be similarly abundant in both undisturbed and restored sites, but absent or very low in occurrence at the disturbed site.’

6. Lines 39-40: Since niche availability was not measured directly, it is speculative to say that your results show hydrologic restoration can enhance niche availability.

7. Lines 47-49: The statement that mangrove forests provide nursery habitat should be referenced.

8. Lines 69-71: Does the Mohd-Azlan, Noske & Lawes, 2015 paper truly support the statement the most mangrove forests have experienced a loss of connectivity and decrease in heterogeneity? I think that this study was limited to a small region in the northern Australia.

9. Lines 79-81: These statements should be referenced.

10. Line 99: Although it may be true that avian assemblages have rarely been used to evaluate restoration success, I would suggest referencing those studies that have, including (but not limited to): Li et al. 2011 ‘Patterns of waterbird community composition across a natural and restored wetland landscape mosaic, Yellow River Delta, China’, Zou et al. 2014 ‘Impact of coastal wetland restoration strategies in the Chongming Dongtan Wetlands, China: Waterbird community composition as an indicator’, Weller 1995, ‘Use of Two Waterbird Guilds as Evaluation Tools for the Kissimmee River Restoration’.

11. Lines 122-130: Similar to my comment on lines 26-29, I would suggest simplifying the objective to either: 1) determine the influence of hydrological restoration on avian diversity and abundance or 2) use avian diversity and abundance as indicators of restoration success. Once the objective is clear, the paper can be framed accordingly.

12. Lines 142-144: This is an incomplete sentence.

13. Lines 213-274: I would also suggest either reducing the number of seemingly redundant analyses i.e. RDAs and GLMs, SIMPER and IndVal, or justify why both are needed.

14. Lines 268-272: What information did you use to group bird species into foraging groups? The sources for this information should be referenced.

15. Lines 310-311: Ardeidae and Threskiornithidae are not species.

16. Lines 318-320: Please justify further why this is a reason for not excluding flying birds.

17. Lines 370-372: It is not clear from the figures presented that this is the main finding of the study. I would suggest providing a figure that shows the relative abundance of bird guilds in restored, disturbed, and undisturbed forests.

Figures

18. Figure 2: I would suggest removing the following results statement from the figure caption: ‘Lack of saturation in the curves indicate that additional effort is required to record the true species richness in the study area.’

19. Figure 3. In the figure caption, please be more specific as to what is meant by ‘only shown the top six’ – are the ‘top six’ the bird species with the highest abundances?

Tables

20. Table 1: Please state in the table caption if the mean values being tested are pooled across years, or are following restoration only. Also, what was the sample size in each group, and where are the post-hoc comparisons reported?

21. Table 3: Please describe in the Table caption what a ‘-‘ indicates in the table.

---

## Round 0.3 · Minor Revisions

After carefully reading the entire manuscript, I am concerned that the term “relative abundance” has been used inappropriately throughout the manuscript. Please carefully check this and revise wording in the text, figures, and tables. I have provided more details below, however because the term was used so many times, I did not point out each problematic instance. Also, as pointed out below, I do think it could be valuable to look at relative abundances among habitat types, however I could not determine whether whether this was done. The figures seems to be plotting abundance rather than relative abundance.

L 32 The term “relative abundance” does not make sense here. Relative abundance is not "the number of individual specimens of an animal or plant seen over a period in a specific place.". Rather, it is specifically defined by ecologists as the “percent composition of an organism of a particular kind relative to the total number of organisms in an area”. I am guessing that you mean to say “abundance” rather than “relative abundance” throughout the text but this is not absolutely clear to me.

L34 After restoration, values of frequency of flooding, water temperature, tree density, and the number of tree species were more similar to that of the undisturbed site than to the values of the disturbed one. – At the restored site, frequency of flooding, water temperature, tree density, and the number of tree species were, more similar to the undisturbed site than to the disturbed site.

L 205 deal with the detectability –help deal with varying detectability

L 331 You seem to be confusing relative abundance with abundance here. Or, do you mean to say “The distribution of species relative abundances and evenness of species distribution were similar...” ? If the distribution of species relative abundances is similar among sites then by definition evenness will be similar among sites.

L 332 “Although the restored site showed higher bird relative abundance” - I don’t see how it is possible to use the term “relative abundance” here. Relative abundance is the percent composition of an organism of a particular kind relative to the total number of organisms in an area. It might be that the restored site has a higher abundance (absolute bird count) and this would be important to know. You could also say that relative abundance of a particular SPECIES was higher in the restored site but it is impossible for all bird species to have higher relative abundance in the restored site (as implied by your sentence) because relative abundance standardizes by the total bird count found in the restored site.

L344 347, 355 etc. It is not clear to me whether you are analyzing abundance or relative abundance here. The definition you gave in your letter for relative abundance is not correct; your definition "the number of individual specimens of an animal or plant seen over a period in a specific place." is simply abundance, not relative abundance. Relative abundance is measured as a percent or proportion and relative abundance always sums to 1 (or 100%) across all species in the area of interest. Wording need to be checked throughout the manuscript when reporting findings related to abundance. A unit such as “number of birds per square meter” or “number of birds per transect” is not referred to as relative abundance; these units could be formally referred to as a “bird density” but they are more commonly referred to simply as measures of “abundance”.

L 393 as posted by – as posited by
L 397 which helped – which probably helped
L 403 by increasing the niche for – by creating niches for
L405 showed higher relative bird abundance – I think this should say “bird abundances” or “bird densities”
L414 “increasing their heterogeneity” – I didn’t understand the reasoning in this sentence. The undisturbed patches were defined in a specific way an the heterogeneity of undisturbed patches is not affected by areas outside the patches. Furthermore, undisturbed patches apparently had the lowest diversity, so how are surrounding patches accounting for low diversity in undisturbed patches?
L 443 – provided better – provide better
L459 “Thus, estimates of relative abundance might be a better indicator.” – I agree with this statement. However, I have not seen any clear estimates of relative abundance presented in this manuscript (see comments on figures and tables below). Relative abundance is calculated as a fraction (% value). The relative abundance of bird A in the disturbed site is calculated as: [total count of bird A in disturbed site] / [total count of all birds in the disturbed site]
L 477 big groups – large groups
L 479 used approaches – approaches used
L483 to what is used – to differing information used
L 529 bringing tidal periodicity – making tidal periodicity

Figure 2 – provide units on the y-axis. I assumed this must be “%” but it looks like they sum to more than 100%, for each condition, which means that the units cannot be relative abundance. It these values are instead counts of individuals then the Y-axis should be labeled as “Abundance” with the units given in parentheses on the axis. Why does the key say “condition” while in Figure 3 it says “habitat”? Why do the legend words sometimes have capital letters and sometimes lowercase? Be consistent.

Figure 3 – same comment as Figure 2 regarding the Y-axis. Regarding the X-axis, if this represents the group rank as indicated, then it is incorrectly plotted. For each habitat, the group rank #1 must be the group type with the highest abundance. Whereas, for example for disturbed habitat, the meateaters should be functional group rank #1 but it is plotted as functional group rank #2. Furthermore, I don’t think this type of graph is effective for conveying these results. It would be much more effective to use a bar graph with abundance plotted on the y-axis and functional groups on the x-axis. Each functional group would have three bars side-by-side, one for disturbed, one for restored, and one for undisturbed.

Table 3 – is it abundance or relative abundance? I doubt relative abundance would meet assumptions of standard glm because the data are proportions.

---

## Round 0.4 · accepted · Accept

Requested corrections have been made I found a few remaining minor corrections as follows; please check while in production:

L 34 were, more similar – were more similar
L 105 second biggest – second largest
L 317 there is a clear dominance – there was a clear dominance
L 419 include another type – include other types
L445 wading and diving, – wading and diving species,
L 508 after of hydrological – after hydrological
L537 To José Nava – We thank José Nava